# Study on the Influence of PCA Pre-Treatment on Pig Face Identification with Random Forest

**DOI:** 10.3390/ani13091555

**Published:** 2023-05-06

**Authors:** Hongwen Yan, Songrui Cai, Erhao Li, Jianyu Liu, Zhiwei Hu, Qiangsheng Li, Huiting Wang

**Affiliations:** 1College of Information Science and Engineering, Shanxi Agricultural University, Jinzhong 030801, China; 2Science & Technology Information and Strategy Research Center of Shanxi, Taiyuan 030024, China

**Keywords:** RF, PCA, individual identification, intelligent management of pig breeding

## Abstract

**Simple Summary:**

Pig face recognition plays an important role in the intelligent breeding and accurate management of pigs, and the mobile and embedded applications of this technology in the management of small and medium-sized pig farms are in great demand; therefore, in order to make the model more suitable for use in small and medium-sized pig farms, in this study, PCA pre-treatment was added to the traditional method. The experiment shows that the model is suitable for small and medium-sized pig farms, and it can promote the intelligent transformation of pig breeding management.

**Abstract:**

To explore the application of a traditional machine learning model in the intelligent management of pigs, in this paper, the influence of PCA pre-treatment on pig face identification with RF is studied. By this testing method, the parameters of two testing schemes, one adopting RF alone and the other adopting RF + PCA, were determined to be 65 and 70, respectively. With individual identification tests carried out on 10 pigs, accuracy, recall, and f1-score were increased by 2.66, 2.76, and 2.81 percentage points, respectively. Except for the slight increase in training time, the test time was reduced to 75% of the old scheme, and the efficiency of the optimized scheme was greatly improved. It indicates that PCA pre-treatment positively improved the efficiency of individual pig identification with RF. Furthermore, it provides experimental support for the mobile terminals and the embedded application of RF classifiers.

## 1. Introduction

As a trend in the development of the pig breeding industry, intelligent breeding and precise management of pigs promote the research and development of relative technologies. During the process, research focus groups in the fields of individual identification, automatic inspection, detection of food and drinking water, gesture measuring, walking behavior detection, and abnormal behavior detection of pigs have been formed, among which individual identification and behavior identification of pigs are the basis for many studies. So far, the research techniques for individual identification and behavior identification of pigs can be classified into three schools. In the early stage, RFID technology [1] was mainly applied to study the efficiency of readers and construct the traceability system for the pig industry, which played a huge role in pig automatic feeding, pork quality management, slaughter supervision, and safety production. As new technologies were developed, the cost of RFID technology was still high, which is why its research and application declined. The second school focuses on the application and research of traditional machine learning models. Many scholars have studied the application of models such as SVM [2,3], KMEANS [4], KNN [5,6], LDA, RANDOMFOREST [7,8], and others and how they can be improved in the field of intelligent management of pigs. Substantial research achievements have been made in individual identification, huddling to keep warm, fast walking and slow walking behavior, statistical calculation of exercise amount, statistical calculation of sleep, and alike. The third school refers to the computer vision technology that mainly focuses on application and improvement of deep learning models. Scholars have used AlexNet, GoogleNet [9], VGG series, YOLO series [10,11,12,13,14], Transformer, and other models to carry out research on individual identification, feeding of food and drinking water, climbing and crossing behavior, and aggressive behavior. An enormous hashrate is needed by this school. At the moment, it is mostly used in laboratories. In the management of small and medium-sized pig farms, mobile terminals and embedded applications are in great need of intelligence technologies. To promote the intelligent development of small and medium-sized pig farms, it is necessary to provide technology and equipment that suits their intelligent management. The application of the first school is limited in the field, while the third school needs a large enough budget to support its hashrate and to have the equipment deployed, which means that it is difficult to spread in small and medium-sized pig farms. The second school, whose identification accuracy and operation efficiency can be better improved, becomes the first choice for the intelligent development of small and medium-sized pig farms due to its low cost as well as abundant research bases.

Therefore, this study explored the influence of PCA pre-treatment on pig face identification with RF. With RF + PCA pre-treatment, though identification accuracy was lost to some degree, training time and testing time decreased to 4.8% and 9% of the original values, respectively. Taking all these factors into consideration, after going through PCA pre-treatment, with the identification accuracy maintained in a range as a premise, the operating efficiency of the RF model improved greatly; thus, it is suitable for the intelligent management technologies to be applied in mobile terminals and embedded application in the management of small and medium-sized pig farms.

Some scholars designed a high-frequency radio frequency identification (HFRFID) system which was validated for its suitability to register individual pigs’ feeding patterns at a round trough in a group-housing context; this system showed good potential for measuring the feeding patterns of growing–finishing pigs in commercial pig houses [15]. Maselyne et al. [16] designed a high-frequency radio frequency identification (HFRFID) system to record the drinking behavior of individual pigs. The sensitivity of the system was 92%, the specificity was 93%, the precision was 90%, and the accuracy was 93%. It improves the productivity and economics of the swine industry as well as the health and welfare of the pigs. Kapun et al. [17] used the UHFRFID system to record the daily activity patterns of pigs. The sensitivity (true positive rate) of the UHFRFID system was about 80% at the feeding trough and playback equipment and about 60% at the drinker. The experimental results show that the system can record the time of pig visits and has a higher data density than video or direct observation. Maselyne et al. [18] proposed a method for constructing feeding visits based on the RFID registration of growing pigs in feeding tanks. The experimental results showed that when two tags were used for each pig, the average sensitivity of the method was 83%, the specificity was 98%, the precision was 97%, the accuracy was 75%, and it can automatically and accurately record the feeding information of growing pigs. Zhu Jun et al. [19] used RFID, intelligent control, network transmission, and other technologies to build a digital breeding pig breeding platform. The platform realizes the integration of automatic fine feeding, environmental control in the pig house, production management, and visual monitoring. Zhu Weixing et al. [20] designed an embedded pig behavior monitoring system based on RFID technology and ARM-Linux. The system is based on the ARM-LINUX platform, and the embedded device is installed in the pig breeding area to monitor the pigs and collect feeding data throughout the day. The experimental results show that the system has good real-time performance and stability. Chen T. Y. et al. [21] used RFID technology to collect pig diet information in the feed area of the pigsty. The experimental results showed that the sensitivity of this technology was 71.1%, the singularity was 87.1%, and the accuracy was 88.8%, realizing the remote monitoring of pig breeding and greatly reducing the labor cost for farmers. LASSO regression and a random forest model were used to predict the weight of pigs at 159 days to 166 days under four scenarios [22]; random forest and generalized linear regression were used to predict the physiological temperature of piglets. However, the prediction error was relatively high [23]; an auto-regression (AR) model and improved local linear embedding (LLE) were used to estimate pig weight in an actual farm environment [24], and SSD [25], F-rcnn [26], and other models were used to study the posture changes of pigs and then analyze their behavior. Win et al. [27] established an automatic pig size sorting system based on computer vision, which can realize personalized feeding according to pig size. Yan Hongwen [28] used Feature Pyramid Attention (FPA) combined with a Tiny-YOLO model to achieve multi-target detection of pigs in different scene groups, and Yan Hongwen [29], based on YOLOV3 as the basic model, introduced spatial attention and channel attention information to construct corresponding attention sub-models and realized the detection of different types of facial poses of group breeding pigs. Kim et al. [30] presented an algorithm that utilizes the major and minor axes of the pig detection box to associate the pig’s head with its corresponding body. They evaluated the detection performance of the YOLOv5 model with respect to the anchor box and demonstrated that the proposed method outperforms the previous method. Hu Zhiwei [31] used ResNet50 and ResNet101 as the backbone network to build a dual attention unit combining channel attention and spatial attention and used it in the feature pyramid network structure, achieving the goal of realizing instance detection of live pigs in different scenarios. Hansen et al. [32] combined the feature extraction results of the VGG-Face model [33] and Fisherfaces with the convolution network constructed by themselves to test a total of 1553 pictures of 10 individual pigs in the natural conditions of the farm and used Grad-CAM [34] activation-resembling mapping to distinguish the adhered pigs, achieving an identification precision rate of 96.7%.

With GPU applied, the high-precision advantage of computer vision technology has gradually been revealed. In the management of small and medium-sized farms, the need for mobile terminals and embedded apps has gradually been increasing. The deep learning model has a high requirement for hardware, and it is difficult to adapt to wide application, while the RFID technology is easy to simulate in the management [35,36] and requires the use of physical tags, such as ear cutting and ear tags, which can easily cause pain to pigs [37]. While it takes time and effort for workers to install tags, it also violates animal welfare breeding values; in addition, low-frequency RFID cannot receive signals from multiple pigs at the same time, and the identification area of high-frequency RFID is very small [38], so RFID technology has been gradually eliminated. Only the hardware requirements of the traditional machine learning model conform to the standards for mobile terminals and embedded apps, though there is still room for improvement in its identification accuracy and running time. PCA can extract the main features of pig faces [39,40], thus reducing the computation burden, improving operation efficiency, eliminating noise interference, and improving identification accuracy.

To promote the application of the traditional machine learning model in mobile terminals and embedded apps, RF is adopted for bettering the pig face identification efficiency in this study, and its influence on the efficiency of pig face identification is further studied by adding PCA pre-treatment, which provides experimental support for its application in both mobile terminals and embedded application. The terms used in this paper are listed in Table 1.

## 2. Materials and Methods

### 2.1. Sample Collection

The data in this study were collected two times. The first time, the data were collected in Dongsongjiazhuang Village, Jicun Town, Fenyang City, Shanxi Province, China (111°95′ E, 37°27′ N). In order to obtain live pig images of different pig house scenes, data were collected from 9:00 to 14:00 on June 1, 2019 (fine, strong light). Three pig farms were selected for video capture; each pig farm consisted of 10–30 pig pens, the number of pigs in each pen varied from 6 to 8, and the size of the pig pens was about 3.5 m × 2.5 m × 1 m. A total of 35 videos of 5 pens of breeding pigs aged 20 to 105 days were collected. The second time, data were collected from 10:30 to 12:00 on 13 October 2019 (cloudy, weak light), and the collection site was located in the Laboratory Animal Management Center of Shanxi Agricultural University, Taigu City, Shanxi Province, China (112°59′ E, 37°43′ N). A total of 15 pigs in 6 pens were selected for video collection. In this study, 10 pigs were selected as the research objects, as shown in Figure 1, including 768 training samples, 85 validation samples, and 250 test samples.

The computer used in the experiment is configured with 64-bit Windows system, Intel Core i7-6700, 8 GB memory, 6 GB video memory capacity, and program development uses Python V3.5 version language.

### 2.2. Principle of Pig Face Identification with RF

In order to ensure that the machine learning model finds the optimal result in each iteration, some random processes are added to most of the machine learning methods. Random forest (RF) [41] also adopts this design idea to construct a random decision tree. For each iteration, the algorithm usually generates the optimal predictive variable [42], with the basic idea of finding the average value of noise to construct sets of decision trees. Through a complex interaction tree, the decision tree in RF maps the complex input space to a simpler space. In reference [28], it is shown that sets of decision trees in RF are randomly trained, and RF reduces over-fitting through the effective use of data. Therefore, the method is an extension of the bagging classification tree, and it is a parallel learning process with fast training and high accuracy as its features. Its advantages are as follows [43]:(1)RF has both anti-over-fitting and anti-noise performances because random steps are included;(2)High-dimensional data can be processed;(3)Learning can be achieved in a parallel way;(4)The training time can be shortened.

With random selection methods adopted, RF performs better when there are many redundant features that cannot be distinguished [44,45]. In this study, RF was used for pig face identification. The complex pig face images were mapped to the category label space via the RF model. The steps for pig face identification with RF are as follows:(1)The n_tree training set is generated by sampling the pig face training samples for n_tree times, with m samples taken each time, for which the Bootstrapping method, a random sampling with replacement method, is used.(2)Every training set needs to train a decision tree model.(3)When splitting the decision tree according to the information gain or gini index, it is necessary to select an optimal feature among all the features.(4)Each decision tree is split in this way, and finally, all the training samples of this node are classified into the same category, and there is no pruning operation in this process.(5)In the end, multiple decision trees will be formed to generate the random forest. In the case of multi-classification tasks, the output of the random forest will be determined by voting.

In this study, 10 different pigs are classified with RF. When each decision tree selects the optimal feature, all the features in the attribute set are read and the feature with the minimum gini coefficient or maximum information gain is selected as the classification standard. Similarly to entropy, the gini coefficient here reflects the uncertainty of the data set. The process of determining is a process of entropy reduction. The gini coefficient is calculated according to Formula (1):(1)$$Gini\left(D\right)=\underset{k}{\sum }\frac{\left|C_{k}\right|}{\left|D\right|}\left(1-\frac{\left|C_{k}\right|}{\left|D\right|}\right)=1-\underset{k}{\sum }{\left(\frac{\left|C_{k}\right|}{\left|D\right|}\right)}^{2}$$
where D represents data collection, [a];

k represents the category label, [dimensionless];

$C_{k}$ represents the kth sample subset, [dimensionless];

$\left|C_{k}\right|$ represents the number of samples included in the kth sample subset, [a].

When feature *A* is selected, the data set D is divided into subset $D_{1}$ and subset $D_{2}$ according to whether the value of feature *A* is a certain eigenvalue, as shown in the following formula:(2)$$D_{1}=\left\{D|A=a\right\}$$
(3)$$D_{2}=\left\{D|A\neq a\right\}$$
where a represents a certain characteristic value of characteristic *A*, [dimensionless];

$D_{i}$ represents the *i*th subset of data set *D*, [dimensionless].

In the case of feature *A*, the gini coefficient of data set *D* is calculated as shown in the formula below:(4)$$Gini\left(D,A\right)=\frac{\left|C_{k}\right|}{\left|D\right|}Gini\left(D_{1}\right)+\frac{\left|C_{k}\right|}{\left|D\right|}Gini\left(D_{2}\right)$$

If the formula listed above achieves the minimum value, then feature *A* is selected; information gain represents the change degree of entropy, i.e., the pre-classification information entropy minus the post-classification information entropy, as shown in Formula (5):(5)$$g\left(D,A\right)=H\left(D\right)-H\left(\left(D|A\right)\right)$$
where $H\left(D\right)$ represents the entropy of the data set *D*, [dimensionless];

$H\left(\left(D|A\right)\right)$ represents the entropy of data set *D* divided by feature *A*, [dimensionless].

Each base learner classifies the pig samples according to the selected feature sequence, while the random forest algorithm uses the absolute majority voting method to make the final classification decision. The rules of the voting method are as follows:(6)$$H\left(x\right)=\left\{\begin{array}{c} \;\;c_{j},\;\;\;if\overset{T}{\underset{i=1}{\sum }}h_{i}^{j}\left(x\right)>0.5\overset{N}{\underset{k=1}{\sum }}\overset{T}{\underset{i=1}{\sum }}h_{i}^{j}\left(x\right);\\ reject,\;\;otherwise. \end{array}\right.$$
where $h_{i}$ represents base learner, [a];

$c_{j}$ represents the category label, [dimensionless];

T represents the number of samples to be tested, [a];

k represents the number of base classifiers, [dimensionless];

j represents the output of the base learner on the category label $\;c_{j}$, [dimensionless].

According to Formula (6), if the random forest has more than half of the predictions for a certain category voted, the final prediction will be this category; otherwise, the prediction will be rejected.

## 3. Comparison Process in the Experiment

### 3.1. Pig Face Identification Test Carried out with Random Forest Alone

#### 3.1.1. Random Forest Model Parameter Determination

In order to determine the number of decision trees and parameters for the splitting quality performance function in the random forest, in this study, multiple tests were designed for the relationship between the number of decision trees, splitting quality performance function model and accuracy, recall rate, as well as f1-score when the pig face images were classified. Splitting quality function included gini and entropy. The number of decision trees was within the range of 0~100. The effects of each parameter combination on the performance of random forest are as shown in Figure 2.

Figure 2a shows the relationship between the number of decision trees within the random forest and the accuracy. The abscissa represents the number of decision trees and the ordinate represents the accuracy. The blue curve represents the gini impurity level “gini” and the green curve represents the information gain “entropy”. It did not matter if it was “gini” or “entropy” that was selected by splitting quality performance function; as the number of decision trees within the random forest increased, there was an obvious growth in the accuracy of forest classification. When the number of decision trees was less than 20, the accuracy grew rapidly; when the number of decision trees was over 20, the accuracy grew slowly; when the number of decision trees was over 65, the accuracy of random forest classification reached 90%; and with the increase in the number of decision trees, the classification accuracy kept stable. The more decision trees there are, the higher the computational complexity of the random forest; in this study, the number of decision trees in the forest was taken as 65, which could not only make the accuracy reach the expected requirement, but also make the computational complexity of random forest relatively low. As can be seen in Figure 2a, “gini” and “entropy” have a similar influence on the accuracy of the model. Therefore, for the splitting quality function of the random forest, either “gini” or “entropy” can be selected.

Figure 2b shows the relationship between the number of decision trees in the random forest and the recall rate. The abscissa represents the number of decision trees, and the ordinate represents the recall rate. Similarly to Figure 2a, the number of decision trees and the recall rate also show a logarithmic growth trend. Since the blue curve fluctuates less than the green recall curve, for the splitting quality performance function of random forest, adopting “gini” would be more stable than adopting “entropy”.

Figure 2c shows the relationship between the number of decision trees in the random forest and the f1-score. The abscissa represents the number of decision trees, and the ordinate represents the f1-score. The number of decision trees and the f1-score also showed a logarithmic growth trend. For the model with gini selected as the parameter, in most cases, the classification performance would be better than entropy. Considering the performance of the three indicators with different amounts of decision trees, the number of decision trees, which would be a parameter for subsequent tests, was taken as 65.

#### 3.1.2. Evaluation Index of RF Model

According to the parameter determined in the above tests, the splitting quality function in the random forest was set as “gini”, and the number of decision trees was set as 65. Tests were conducted on the test set, and according to the test results, the confusion matrix was drawn, as shown in Figure 3.

The leftmost column of the confusion matrix represents the real category, the top row represents the predicted category, and the diagonal line represents the number of correct predictions. The *precision*, *recall*, and *f*1*-score* values of ten different pigs were obtained in accordance with the confusion matrix and Formulas (7)–(9), respectively, as shown in Table 2. The precision ratio was defined as
(7)$$precision=\frac{TP}{TP+FP}$$

The recall ratio was defined as
(8)$$recall=\frac{TP}{TP+FN}$$

The recall ratio was also called the recall rate. The recall ratio and precision ratio changed in an opposite trend. The *f*1*-score* can measure the different preferences of these two indexes, and the formula was as follows:(9)$$f1-score=2\times \frac{precision\times recall}{precision+recall}$$
where

*TP* represents the number of positive samples that are actually positive samples, [a];

*FP* represents the number of positive samples that are actually negative samples, [a];

*FN* represents the number of negative samples that are actually positive samples, [a].

As can be seen from Table 2, the average accuracy of classification and identification of pig face data with RF adopted reached 90.61%, with the recall rate and the f1-score reaching 89.76% and 89.79%, respectively.

### 3.2. Experiment of Pig Face Identification with RF + PCA Pre-Treatment

#### 3.2.1. Determination of the k Value in Principal Component Analysis

At the first stage of the experiment, the number of principal components needs to be determined for principal component analysis. Here, the k value was taken as 300, with the variance explanation rate reaching over 95%.

#### 3.2.2. Determination of RF Parameters in the Optimization Plan

The distribution of the data going through PCA dimension reduction may change to some extent; thus, the number of decision trees determined in the previous stage may not be the optimal value when used as the input for the RF model. Therefore, the number of decision trees in the RF model needs to be redetermined. With the same testing method adopted as in Section 3.1.1 ‘Random Forest Model Parameter Determination’, the relationships between the number of decision trees and the classification accuracy, recall rate, as well as f1-score of the RF model to pig face data were measured, and their relationship is as shown in Figure 4.

In Figure 4a, the abscissa represents the number of decision trees in the random forest, and the ordinate represents the precision. The green broken line represents entropy, and the blue broken line represents gini. When the number of decision trees was 1, the precision value was around 0.45, which meant that the classification effect was even lower than random guess. As the number of decision trees increased, the corresponding precision values grew rapidly. When the number of decision trees reached about 20, the precision value reached above 0.85. Then, as the number of decision trees continued to increase, the corresponding precision values grew slowly, accompanied by a small amplitude of oscillation. The overall performance of the green broken line, whose precision value reached the maximum of around 0.92 at 70, was better than that of the blue broken line. For the classification performance of the model, the accuracy, recall rate, and f1-score value need to be considered in a comprehensive way. According to the test results, the broken-line graphs that represent the relationship between the recall rate and the f1-score value for different numbers of decision trees were drawn, as shown in Figure 4b,c.

As shown in Figure 4b,c, if entropy was chosen as the parameter for the model, its performance would be better than gini. As the number of decision trees increased, entropy showed the same trend as precision, reaching a maximum of 0.92 and 0.90 when the number of decision trees was around 70. With the classification performance index for different numbers of decision trees considered, the optimal classification results were obtained when the number was 70.

#### 3.2.3. Model Evaluation Index of Optimization Plan

The parameter determined in Section 3.2.2 ‘Determination of RF Parameters in the Optimization Plan’, “entropy”, was used for the splitting quality function in the random forest, and the number of decision trees was taken as 70. Tests were carried out on the test set, and the confusion matrix was drawn, as shown in Figure 5, in which the left-most column of the matrix refers to the actual category, the top row refers to the category in the actual prediction, while the leading diagonal shows the number correctly predicted, and the right-most column is a visual representation of the correct values for each category participating in the prediction.

The precision, recall, and f1-core values of 10 different pigs were obtained in accordance with the confusion matrix and Formulas (7)–(9), as shown in Table 3.

The RF classifier with PCA pre-treatment adopted was used for individual pig identification, which not only improved the identification accuracy but also reduced the testing time of the model. The specific test indexes for the two schemes are as shown in Table 4.

It can be seen from Table 4 that, with the optimization plan of PCA + RF pre-processing, except for the slight increase in training time, the other evaluation indicators have improved. In order to study the effect of the PCA preprocessing method on the efficiency of the machine learning model in pig individual recognition, the model running efficiency of pig individual recognition using SVM, KNN, PCA + SVM, and PCA + KNN is simultaneously compared through experiments. The results are shown in Table 5.

According to Table 5, the accuracy of the SVM, KNN, and RF models reached 83.66%, 91.46%, and 90.61%, respectively, before PCA pre-treatment; the accuracy of PCA + RF was the highest (93.22%), and the training time of the three models increased or decreased in the aspect of running efficiency; however, the testing time directly related to the practical application is reduced. Although the PCA + SVM and PCA + KNN methods reduce greatly, their accuracy is only 82.82%; PCA + RF is the main content of further research.

In addition, in other studies [28,29,31,46,47,48,49,50,51], the authors studied the role of modern neural networks, such as YOLO, AlexNet, and Tiny-YOLO, in pig identification, head and face postures, and behavioral analysis. However, the modern neural network has a large number of parameters and a deep hierarchical structure, which has become the bottleneck of its research results in the development of embedded applications; this is one of the starting points of this study.

## 4. Discussion

As can be seen in Table 4, in PCA + random forest optimizing test scheme, the precision reached 93.22%, the recall value was 92.52%, and the f1-score was 92.60%, which increased by 2.66, 2.76, and 2.81 percentage points, respectively, compared with the values obtained in the random forest alone scheme. In terms of the operating efficiency of the algorithm, the training time increased slightly from 1229 ms to 1340 ms, with an increase of 9%, though the testing time decreased from 8 ms to 6 ms. In the new scheme, only 75% of the testing time for the old scheme is needed. This improvement better suits actual production scenarios.

The accuracy, recall rate, and f1-score improved for the following two reasons: First, after the original pig sample data underwent PCA feature extraction, the model had the most identifiable features of the pigs extracted, which was conducive to the improvement of accuracy. Second, with PCA employed, the secondary features were ignored, and the noise in the data was filtered; thus, the model possessed better generalization ability, as was verified in the tests for the validation set. PCA needed some time to process, and the training time was increased, while the post-optimization testing time was greatly improved and even reduced to 75% of that of the old scheme. With the PCA + random forest test scheme used, the identification accuracy of the algorithm was improved; the operating efficiency also improved, which provides both theoretical and experimental support for further embedded application of the algorithm.

## 5. Conclusions

This paper studied the influence of the PCA pre-treatment method on the efficiency of identifying ten pigs with an RF classifier. The parameters of the classifier were determined through tests. By comparing the influence of the two testing schemes, in which one adopted the RF classifier alone and the other adopted PCA + RF, on identification efficiency, the following conclusions are drawn:(1)For individual identification of pigs, the RF classifier can be used, for which the parameter selection is relative to the pre-treatment method. If RF alone is used, the splitting quality function shall be “gini”, and the number of decision trees shall be 65; in the case of the PCA + RF optimization scheme, the corresponding parameters shall be “entropy” and 70.(2)PCA pre-treatment can increase the efficiency of individual pig identification with RF, and the accuracy, recall rate, and the f1-score are increased by 2.66, 2.76, and 2.81 percentage points, respectively, while the testing time is reduced to 75% of the original value.(3)The RF classifier that underwent PCA pre-treatment is more suitable for application in mobile terminals and embedded application, and it is suitable for the development of a portable and real-time pig face identification system; thus, the cost of intelligent breeding and management of small and medium-sized farms can be reduced, and the process of intellectualization of small and medium-sized farms can be promoted.

## Figures and Tables

**Figure 1 animals-13-01555-f001:**
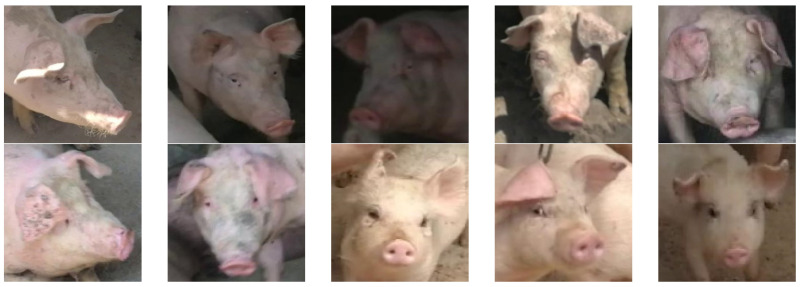
Pig Samples.

**Figure 2 animals-13-01555-f002:**
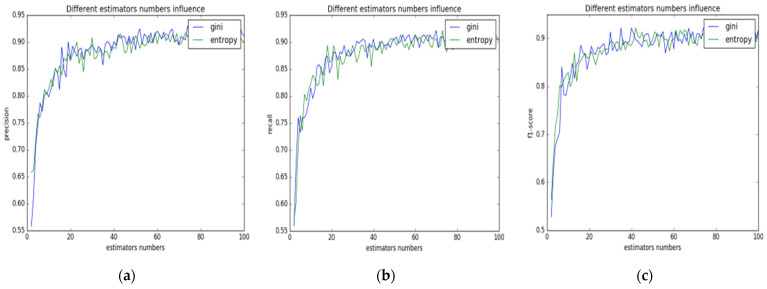
The relationship between the evaluation index of RF model and its parameter. (**a**) Accuracy; (**b**) recall rate; (**c**) f1 value.

**Figure 3 animals-13-01555-f003:**
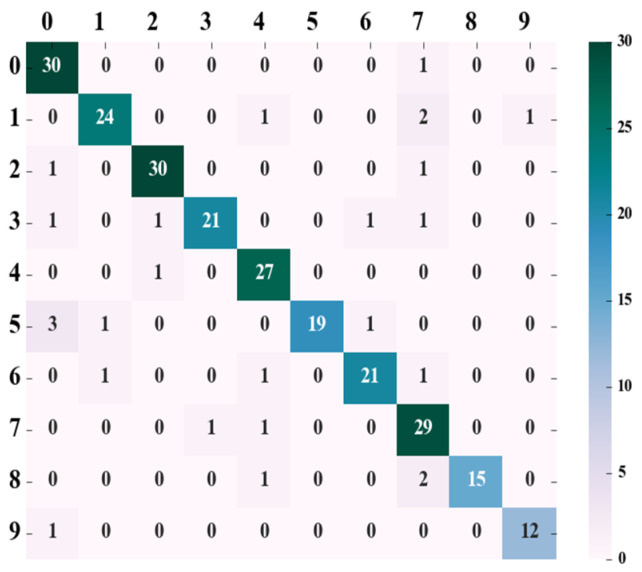
RF prediction result confusion matrix.

**Figure 4 animals-13-01555-f004:**
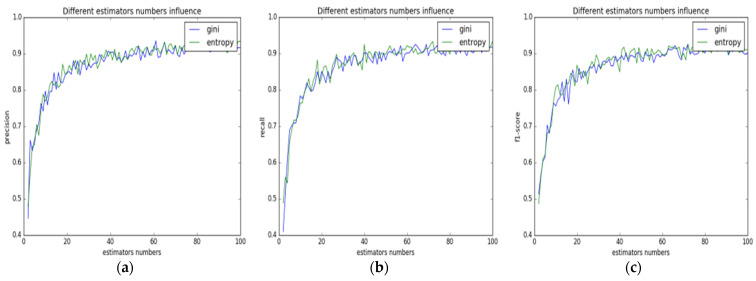
The relationship between RF model performance and parameters after pre-PCA processing. (**a**) Accuracy; (**b**) recall rate; (**c**) f1 value.

**Figure 5 animals-13-01555-f005:**
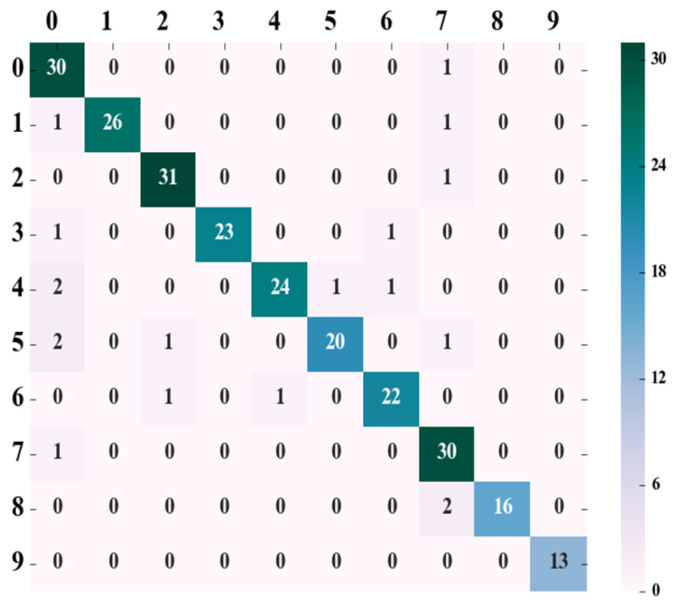
RF + PCA prediction result confusion matrix.

**Table 1 animals-13-01555-t001:** Named terms list.

Abbreviation	Meaning
PCA	Principal Component Analysis
RF	Random Forest
RFID	Radio Frequency IDentification
HFRFID	High-Frequency Radio Frequency IDentification
UHFRFID	Ultra-High Frequency Radio Frequency IDentification
SVM	Support Vector Machine
KNN	K-Nearest Neighbors
LDA	Linear Discriminant Analysis
VGG	Visual Geometry Group
SSD	Single Shot Detector
F-rcnn	Faster Region-based Convolutional Neural Networks
YOLO	You Only Look Once

**Table 2 animals-13-01555-t002:** RF model prediction performance table.

Category	Precision (%)	Recall (%)	f1-Score (%)	Count [a]
1	72	100	84	31
2	96	89	93	28
3	97	91	94	32
4	95	84	89	25
5	87	96	92	28
6	100	79	88	24
7	91	88	89	24
8	78	94	85	31
9	100	83	91	18
10	92	92	92	13
average	90.61	89.76	89.79	25

**Table 3 animals-13-01555-t003:** RF + PCA model prediction performance table.

Category	Precision (%)	Recall (%)	f1-Score (%)	Count [a]
1	81	97	88	31
2	100	93	96	28
3	94	97	95	32
4	100	92	96	25
5	96	86	91	28
6	95	83	89	24
7	92	92	92	24
8	83	97	90	31
9	100	89	94	18
10	100	100	100	13
average	93.22	92.52	92.60	25

**Table 4 animals-13-01555-t004:** The optimization result of the RF model by PCA pre-processing.

Model	Precision (%)	Precision Change	Test_Time (ms)	Test_New/Old (%)	Train_Time (ms)	Traintest_New/Old (%)
RF	90.61	0	8	100	1229	100
PCA + RF	93.22	+2.61	6	75	1340	109

**Table 5 animals-13-01555-t005:** Comparison of generalization test results.

Model	Precision (%)	Precision Change	Test_Time (ms)	Test_New/Old (%)	Train_Time (ms)	Traintest_New/Old (%)
SVM	83.66	0	329	100	12,823	100
KNN	91.46	0	1306	100	187	100
RF	90.61	0	8	100	1229	100
PCA + SVM	88.85	+5.19	69	20.9	3861	30.1
PCA + KNN	82.82	−8.64	93	7	9	4.8
PCA + RF	93.22	+2.61	6	75	1340	109

## Data Availability

Not applicable.

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
