# Peer review of "Study on the Influence of PCA Pre-Treatment on Pig Face Identification with Random Forest"

_animals, 2023, doi:10.3390/ani13091555_

Round 1

Reviewer 1 Report

The work is interesting

Some suggestions

Implement more machnine lerning techniques for example neyral networks or Bayesian networks, Discriminal analysis and compare them with the methodology inside the paper.

The paper is only an implemenation paper without mathematical analysis. It is needed to include more mathematics

Reviewer 2 Report

Study on the Influence of PCA Pre-treatment on Pig Face Identification with Random Forest

The article's subject is relevant to the pig industry and the intelligent management of pigs. The text approaches the subject with scientific correctness. The article is well organized, and it is simple to follow. Despite this, some improvements can be made to make the text clearer. The Tables and Figures are relevant for understanding the article. The material and methods are clearly described, allowing for a perfect understanding by other researchers. When discussing the results, it is important to provide context by referring to other studies conducted in the same area. By doing so, you can highlight the strengths and weaknesses of your research and demonstrate how your findings contribute to the existing body of knowledge. Finally, the results support the conclusions.

Some detailed comments are below:

L18-adopting RF+PCA, were determined “change with” adopting RF+PCA, was determined

L22-pre-treatment had a positive effect on improving the efficiency “change with” pre-treatment positively improved the efficiency

L22-23 It provides experimental support for the mobile terminals and embedded application of RF classifiers. “change with” Furthermore, it provides experimental support for the mobile terminals and the embedded application of RF classifiers.

L39 still high, that is why its “change with” still high, which is why its

L49 carry out researches on “change with” carry out research on

L51 in the laboratories. “change with” in laboratories.

L52 embedded application are “change with” embedded applications are

L61-62 Therefore, in this study, the influence of PCA pre-treatment on pig face identification with RF was explored. “change with” Therefore, this study explored the influence of PCA pre-treatment on pig face identification with RF.

L78 predict physiological “change with” predict the physiological

L78 of piglets, though the “change with” of piglets. However, the

L82 then analyse their “change with” then analyze their. Please check other inconsistencies between American and British spelling

L104 requirement to hardware, it is difficult “change with” requirement for hardware, and it is difficult

L110-111 thus reduce the computation burden, improve operation efficiency, eliminate noise interference and improve identification accuracy “change with” thus reducing the computation burden, improving operation efficiency, eliminating noise interference and improving identification accuracy.

L119 study was collected in two “change with” study were collected two

153-154 In essence, the complex pig face images was space mapped to the category label space via RF model. “change with” The complex pig face images were mapped to the category label space via the RF model.

L162 way, and finally all “change with” way, and finally, all

L286 some extent, thus the number “change with” some extent; thus the number

L292 relationship are as shown “change with” relationship is as shown

L323 category in actual prediction “change with” category in the actual prediction

L341 points respectively compared “change with” points, respectively compared

L344-345 That is to say, in the new scheme, only 75% of the testing time for the old scheme is needed. “change with” In the new scheme, only 75% of the testing time for the old scheme is needed.

L355 improved, also operating “change with” improved; also operating

L359 In this paper, the influence of PCA pre-treatment method on the efficiency of identification of 10 pigs with RF classifier was studied. “change with” This paper studied the influence of the PCA pre-treatment method on the efficiency of identifying ten pigs with an RF classifier.

Reviewer 3 Report

This paper starts by presenting state-of-the-art techniques for research and development of relative technologies in the pig-breeding industry. In particular, the aim of this study was to study the influence of PCA pre-treatment method on the efficiency of the identification of pigs with an RF classifier.

Overall, the introduction and the materials and methods and their results are quite well done.

However, this reviewer is suggesting that prior to publication, the authors consider the following minor editorial revisions:

About the introduction, I suggest that the authors add citations for the RFID part.

There are many abbreviations, for this reason, I suggest to the authors to report a short table with a short description of each parameter.

The authors, in the future, should also consider adding some papers about the machine learning approach.

The following typographical error was detected:

-        In line 75 allow

-        In line 119 and 153  I suggest were and not was

About the Materials and Methods, I am not qualified to assess the quality of the work.

In general, the manuscript is reasonably well done and close to being in the appropriate form for future publication, so it is recommended that the authors consider some of these minor revisions and resubmit the revised manuscript for future publication.

Round 2

Reviewer 1 Report

Accept as it is